# Effectiveness of a novel midwifery preceptor program on midwives' competence and confidence in Sierra Leone: A prospective pilot study

Brittney J. van de Water[1,2,3]*, Ashley H. Longacre[1], Mustapha Sonnie[4], Mary Moran[1], Jenny Hotchkiss[5], Marian Bangura[4], Isha Beckie Sesay[4], Frances Fornah[6], Patricia Juana-Kamara[7], Chrisensia Owoko[4], Jennina Rose Viera[4], Alice Konyani[4], Adelaide Debrah[4], Mary Augusta Mamakoh Fullah[8], Julie Mann[3]

**1** Connell School of Nursing, Boston College, Chestnut Hill, Massachusetts, United States of America, **2** Schiller Institute for Integrated Science and Society, Boston College, Chestnut Hill, United States of America, **3** Seed Global Health, Boston, Massachusetts, United States of America, **4** Seed Global Health, Freetown, Sierra Leone **5** Morrissy College of Arts and Sciences, Boston College, Chestnut Hill, Massachusetts, United States of America, **6** School of Midwifery Makeni, Makeni, Sierra Leone, **7** School of Midwifery Bo, Bo, Sierra Leone, **8** Ministry of Health and Sanitation, Freetown, Sierra Leone

* brittney.vandewater@bc.edu

## Abstract

Sub-Saharan Africa faces some of the highest maternal mortality rates globally and the greatest shortage of midwives. Strengthening midwifery education and clinical training is one of the most impactful interventions to reduce maternal mortality. Global reports cite lack of investment in educators, limited skills and knowledge in contemporary teaching methods, and limited clinical experience for students as key barriers. To address these gaps, a six-month, low-dose, high frequency midwifery preceptor course was developed and piloted in Sierra Leone. This study aims to evaluate the program's effectiveness in strengthening preceptors' clinical and precepting competence and confidence. Between 2023 and 2024 two midwifery schools in Sierra Leone participated in this novel intervention, each with n = 10 midwife preceptors (N = 20). A pre-post-test design was used to evaluate participants at the end of the intervention (6-months). Twenty-five assessments were used to evaluate participants including: written competency test, objective structured clinical examinations (OSCEs), self-assessments of competence and confidence, direct clinical observation, and student evaluations. Written competency tests had significant statistical improvement from pre- to post-assessment (p < 0.0001). All OSCEs had significant statistical improvement from pre- to post-test. The largest mean difference was for incomplete abortion/manual vacuum aspiration (mean: 45.76, SD: 15.47) while the smallest mean difference was in repair of perineal tears (mean: 9.26, SD: 11.91). Self-assessed clinical confidence and competence were also significant with average

**Data availability statement:** Data have been deposited in the Boston College Dataverse and is available at https://doi.org/10.7910/DVN/T676OX.

**Funding:** This research was supported by a 2022 Innovation Grant by the Boston College William F. Connell School of Nursing (BvdW, AL, MS, JM) and Seed Global Health (BvdW, MS, AK, JV, AD, CO, BS, MB, JM).

**Competing interests:** The authors have declared that no competing interests exist.

**Abbreviations:** ICM, International Confederation of Midwifery; WHO, World Health Organization.

median scores increasing 1.42 points (IQR: 0.86, p = 0.002) and 0.75 points (IQR: 0.77, p = 0.0005) respectively. Self-assessed precepting competence significantly improved with average median scores increasing 1.32 points (IQR 0.80, p = 0.0002). Direct clinical observations had no change and student evaluations improved slightly pre- to post-test. Participants showed statistically significant improvements in clinical and precepting competence and confidence. This midwifery precepting program may be a useful intervention in strengthening midwifery education and maternal health-care in a region with the highest burden of maternal mortality.

## Introduction

Maternal mortality is largely preventable and disproportionately affects sub-Saharan Africa, which accounted for 70% of global maternal deaths in 2020 [1]. Additionally, in 2022, sub-Saharan Africa had the highest neonatal mortality rate in the world at about 27 neonatal deaths per 1,000 live births [2]. Both maternal and neonatal mortality can be prevented with improved quality of care at birth and throughout the days immediately following birth, including strengthening midwifery care [3].

Strengthening midwifery education and the quality of midwifery care is one of the most impactful interventions to address maternal and newborn morbidity and mortality [1,2,4,5]. Multiple global reports have identified that lack of investment in educators, limited skills and knowledge in contemporary teaching and learning, and inadequate "hands-on" experience for students are key barriers to strengthening midwifery education and care [4]. Creating a strong cohort of midwifery preceptors in the clinical setting is one way to address this challenge in midwifery education. A preceptor is an "experienced, actively practicing healthcare provider who provides structured opportunities for students to gain experience in the clinical setting." [6]. Preceptor programs have shown effective ways to promote critical thinking and advance learning of clinical concepts as well as improving patient outcomes [7].

Seed Global Health (Seed) has provided midwifery education in sub-Saharan Africa for over a decade [8]. In 2019, Seed was asked by the government of Sierra Leone to complete a needs assessment in the country consisting of a mixed-methods evaluation of midwifery institutions evaluating student, graduate, preceptor, and maternal health outcomes from various interested parties [9]. One of the main findings was the need for additional preceptors and higher quality preceptors in the clinical setting. Therefore, we developed a comprehensive Midwifery Preceptor Program (MPP) to train practicing midwives to confidently and competently precept students during their clinical rotations in midwifery. This MPP consists of four steps with multiple activities per step (described in detail below). The organizational readiness to implement this change – the MPP – in these low-resourced settings has previously been assessed, and both schools had high change commitment and efficacy; two attributes affecting decisions to adopt educational programs like this MPP which have been used in other sub-Saharan settings [10]. Process maps were also co-created with MPP partners prior to implementing the program to understand modifiable

system challenges at the schools and clinical sites where the intervention will take place [11]. Therefore, the aim of this study is to determine the effectiveness of the first year of the MPP on participants' competence and confidence after the intensive phase of the program.

## Methods

### Ethics statement

Ethical approval for this study was obtained from the Boston College Institutional Review Board (#23.147.01) on January 23, 2023 and the Sierra Leone Ethics and Scientific Review Committee on February 1, 2023 (SLESRC No, 017/02/2023). Data for this study were accessed and entered into REDCap for research purposes beginning September 12, 2023. Research staff had access to participants names on assessment forms during data collection; however, participant names were not entered into REDCap assessment forms. Informed consent was not obtained because both IRBs deemed consent not to be necessary for this aspect of the study. Signed informed consent was obtained from participants for a qualitative aspect of the study published elsewhere [12].

### Study design and Intervention

The launch of the pilot-year of the program took place in February of 2023. This prospective cohort study assesses midwives' clinical competence and confidence at baseline and completion of the intensive phase of the intervention (Step 2) of the Midwifery Preceptor Program. The 40-week preceptor program took place at two schools of midwifery in Sierra Leone. Recruitment began 01/03/2023 and all assessments were complete by 31/05/2024 before a graduation ceremony in May 2024. The MPP consists of four steps, each with multiple activities per step: (1) 4-weeks of partnership building/ program development, (2) a 40-week preceptor course, (3) 4-weeks of training the trainer activities, and (4) 4-weeks of program sustainability activities [13].

The 40-week preceptor course consisted of a 12-week midwifery skill-building module, a 12-week preceptor training module, and ongoing clinical support and supervision in the clinical setting (during the 24- weeks of the modules, and the following 16 weeks). Modules were presented in a low-dose, high-frequency format with weekly one- to two-hour multi-modal sessions (i.e., simulation, didactic, case-based scenarios), followed by facilitated hands-on supervision from MPP facilitators. This low-dose, high-frequency model has been shown to support maternal health interventions across Africa due to repeated sessions, simulation, team cohesion and its cost-effectiveness [14–17]. This program has a low facilitator to preceptor ratio (5:1) to provide consistent and ongoing mentorship. Finally, there was also a 2-day opening retreat, 1-day mid-program retreat, and a closing graduation ceremony with the intent to provide and model self-care behaviors (i.e., mindfulness, yoga, etc) and team building exercises. This intervention differs from many existing training programs due to its low-dose, high-frequency model, low facilitator to preceptor ratio, program duration, and ongoing mentorship on a weekly basis throughout the program both through didactic and hands-on clinical experiential learning [18–20]. The logical assumption underlying this program is that with sufficient inputs and planned activities, certain improved outputs and outcomes, as well as eventual long-term impact will occur (Fig 1) [13].

The curriculum consisted of 25 modules about ICM Essentials for Midwifery. Another 10 modules focused on adult learning theory and principles of precepting. Facilitators in year 1 of this program were all Seed Global Health Educators and had at least a bachelor's or master's degree in midwifery, went through a rigorous application and two-step interview process, had at least five years' experience as a nurse-midwife, and were visiting faculty at the institutions in Sierra Leone where Seed had placed them. As mentioned previously, JM (Associate Director of Midwifery at Seed Global Health) had a bi-weekly meeting to discuss the preceptor program with the facilitators and spoke ad hoc with facilitators regularly and visited sites bi-annually with Seed country staff (MS, MB, IS) visiting at least quarterly.

PLOS Global Public Health

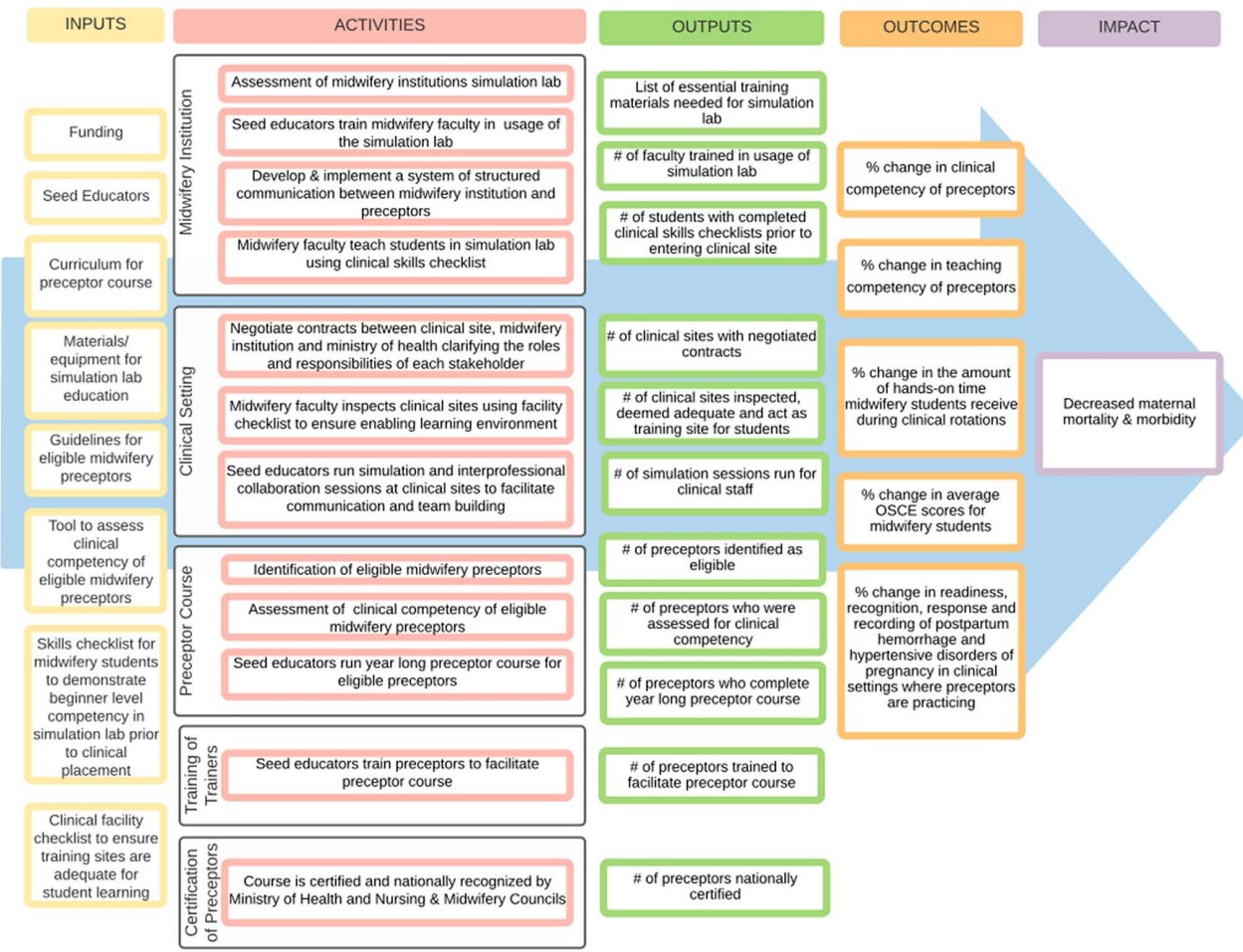

**Fig 1. Preceptor program logic model.** Originally published: BMC Nurs. 2024 May 31;23:365. doi: https://doi.org/10.1186/s12912-024-02036-2.

## Setting

The study occurred in two cities; Bo and Makeni, Sierra Leone. Bo, the second largest city in the country, has Bo Government Hospital, which is a large government-run, regional referral hospital [21]. It has approximately 3,600 births a year. Makeni, the fifth largest city in the country, has Makeni Regional Hospital, which is also a large government-run, regional referral hospital and performs approximately 2,600 births a year [21]. Bo and Makeni both have a public midwifery school with diploma and direct entry programs and a population of approximately 200 students each.

## Population

A total of 20 midwives participated in this preceptor program, ten at each facility (Table 1). Participants had to have a minimum of 2 years full-time clinical experience across the midwifery scope of practice within the last five years, have completed an accredited midwifery program and hold a current midwifery license, complete an application form and submit a letter of recommendation from a supervisor. The average age of participants was 43 years and all were female. Nineteen (95%) participants worked at public institutions, and 90% worked at regional referral hospitals. Participants had a mean of

**Table 1. Demographics of preceptor program participants.**

|  | Site 1 (n = 10) | Site 2 (n = 10) | Total (N = 20) |
|---|---|---|---|
| Age (years); Mean, SD) | 43 (7.78) | 43 (6.73) | 43 (7.04) |
| Female (%) | 100% | 100% | 100% |
| Public institution (%) | 90% | 100% | 95% |
| Regional referral hospital (%) | 90% | 90% | 90% |
| Years of nursing or midwifery experience (Mean, SD) | 11.38 (3.81) | 13.50 (7.41) | 12.56 (6.02) |
| Previous experience in precepting/clinical teaching (%) | 80% | 90% | 85% |

12.56 years of experience as a nurse or midwife, and the majority (85%) had experience in precepting or clinical teaching prior to this program.

## Tools/procedures

We included 25 assessments which were used at baseline and end of the intervention. Eighteen assessments consisted of skills checklist questions assessed on a 0–2 scale, four assessments contained Likert scale questions, and three assessments were multiple choice questions. The assessments covered 18 topics about essential competencies of midwifery practice (see Table 2). OSCE, simulation and direct observation tools were developed using national protocols and evidence-based practice guidelines and were internally reviewed by midwifery faculty in Sierra Leone to ensure application to the local context. Teaching assessment tools were adapted from the American College of Nurse Midwives [22]. Self-assessment of Midwifery Competencies was developed from ICM Essential Competencies of Midwifery Practice [23]. Across the two sites, there were four facilitators who rated participants. The four facilitators met regularly (at least bi-weekly) with the director of midwifery (JM) to discuss grading and ensure internal consistency in scoring assessments.

## Analysis

A pre-post-test design was used to evaluate participants at the end of the intervention (6-months). Thirteen objective structured clinical examinations (OSCEs) and five direct observation checklists were used to determine mean differences between scores. Data were systematically excluded for skills that were not observed during these assessments; thus, no score was imputed or included in the final analysis for any unperformed skills. This approach ensures that the total scores and specific performance metrics accurately reflect only the competencies that were directly evaluated. Paired t-tests were used for each of the OSCEs/observation checklists with a preset alpha of 0.05.

For self-assessments of clinical and precepting confidence and competence, Wilcoxon signed-rank tests examined the difference in median pre- and post-scores with participants answering 3- or 5-item Likert scale questions. Similarly, for student evaluations, students individually answered 4-item Likert scale questions about their preceptor (i.e., the participant in the program); a Wilcoxon signed-rank test was used to examine the difference in those median pre- and post-scores. The Wilcoxon signed-rank test, the non-parametric equivalent to a paired t-test, was selected because the differences between pre- and post-intervention scores were not normally distributed, and is commonly used for Likert item data [24].

Study data were collected and managed using REDCap electronic data capture tools hosted at Boston College. REDCap (Research Electronic Data Capture) is a secure, web-based software platform designed to support data capture for research studies, providing 1) an intuitive interface for validated data capture; 2) audit trails for tracking data manipulation and export procedures; 3) automated export procedures for seamless data downloads to common statistical packages; and 4) procedures for data integration and interoperability with external sources [25,26]. All statistical analyses were performed using SAS software, version 9.4 (SAS Institute Inc., Cary, NC).

**Table 2. List of assessments used in the study.**

| Assessment name | Type of questions (M/C, Likert scale, 0–2 scale) | Assessment Method/Format (OSCE, self-administered, direct observation in clinical setting, peer/student evaluation) | Number of questions in assessment |
|---|---|---|---|
| **Clinical Competency and Confidence** | | | |
| Clinical Competency test | Multiple choice | Written exam | 85 |
| Skills Checklist: Severe Pre-Eclampsia/ Eclampsia | 0-2 Scale | OSCE | 41 |
| Skills Checklist: Incomplete Abortion/Manual Vacuum Aspiration | 0-2 Scale | OSCE | 31 |
| Skills Checklist: Newborn Resuscitation | 0-2 Scale | OSCE | 20 |
| Skills Checklist: Insertion of IUD | 0-2 Scale | OSCE | 25 |
| Skills Checklist: Infection in Labor | 0-2 Scale | OSCE | 24 |
| Skills Checklist: Vaginal Exam in Labor | 0-2 Scale | OSCE | 24 |
| Skills Checklist: Repair of 1st and 2nd Degree Perineal Tears | 0-2 Scale | Direct observation in clinical setting | 21 |
| Skills Checklist: Antenatal History, Vital Signs, and Antenatal Counseling | 0-2 Scale | OSCE | 54 |
| Skills Checklist: Assessment in Labor | 0-2 Scale | OSCE | 42 |
| Skills Checklist: Breech Delivery | 0-2 Scale | OSCE | BO: 19 MAK: 20 |
| Skills Checklist: Examination of the Newborn | 0-2 Scale | OSCE | 23 |
| Skills Checklist: Respectful Maternity Care | 0-2 Scale | Direct observation in the clinical setting | 17 |
| Skills Checklist: Shoulder Dystocia | 0-2 Scale | OSCE | 19 |
| Skills Checklist: Spontaneous Vaginal Delivery | 0-2 Scale | Direct observation in the clinical setting | 43 |
| Skills Checklist: Teamwork & Communication | 0-2 Scale | Direct observation in the clinical setting | 12 |
| Skills Checklist: Augmentation of Labor | 0-2 Scale | OSCE | 34 |
| Skills Checklist: Postpartum Hemorrhage - Atony and Manual Removal | 0-2 Scale | OSCE | 30 |
| Skills Checklist: Using the Partograph | 0-2 Scale | Direct observation in the clinical setting | 21 |
| Self-Assessment of *Clinical* Competence and Confidence | Likert Scale (1–3) | Self-assessment | 237 |
| **Precepting/Teaching Competency and Confidence** | | | |
| Student's Evaluation of Preceptor | Likert Scale (0–3) | Student evaluation | 28 |
| Self-Assessment of *Preceptor's* Competence in Precepting | Likert Scale (0–5) | Self-assessment | 50 |
| Observed Assessment of Preceptor's Competence in Precepting | Likert Scale (0–5) | Direct observation in the clinical setting | 50 |

## Results

### Feasibility

All twenty preceptors who started the program completed and graduated from the program for a 100% completion rate.

### Clinical competency and confidence

**Clinical competency test.** The clinical competency test had statistically significant improvement from pre- to post-assessment. Pre-program, the participant mean score was 64.88 (SD: 8.94) out of 100, and the post-program mean score was 81.13 (SD: 10.16) for a post-pre mean difference of 16.25 points (SD: 9.71, 95% CI: 11.7014, 20.7896, $p < 0.0001$).

**Hands on clinical skills assessment.** *OSCE*

All OSCE scores had statistically significant improvement from pre- to post-assessment (Table 3). The largest mean difference was for incomplete abortion/manual vacuum aspiration (mean: 45.76, SD: 15.47) while the smallest mean difference was for assessment in labor (mean: 13.05, SD: 13.98).

*Direct observation in the clinical setting*

Five competencies of midwifery were assessed via direct observation with patients in the clinical setting. All scores had statistically significant improvement from pre- to post-assessment (Table 4). The largest mean difference was for respectful maternity care (mean: 18.10, SD: 18.56) while the smallest mean difference was in repair of perineal tears (mean: 9.26, SD: 11.91).

**Clinical skills self-assessments.** The clinical skills self-assessment was adapted from the International Confederation of Midwives self-assessment tool that is intended for midwives to self-assess both their competency and confidence in meeting the essential competencies of midwifery across four overarching categories: general competencies; pre-pregnancy and antenatal care; care during labor and birth; ongoing care of women and newborns.

*Confidence*

Overall, preceptor self-assessed confidence in precepting differed significantly with average median scores increasing 1.43 points (IQR: 0.85, p = 0.002) between pre- and post-program (Table 5). Across the four competency categories

**Table 3. OSCE Competencies and mean difference in pre- and post-assessment score.**

| OSCE Competencies | Mean Difference (post - pre) & SD | 95% CI for Mean Difference | p-value | n* |
|---|---|---|---|---|
| Incomplete abortion/manual vacuum aspiration | 45.76 (SD: 15.47) | (38.31, 53.22) | <.0001 | 19 |
| Antenatal history | 32.31 (SD: 13.00) | (26.22, 38.39) | <.0001 | 20 |
| Augmentation of labor | 30.80 (SD: 26.47) | (18.04, 43.56) | <.0001 | 19 |
| Shoulder dystocia | 28.68 (SD: 13.27) | (22.08, 35.28) | <.0001 | 18 |
| Insertion of IUD | 28.09 (SD: 22.88) | (15.90, 40.29) | 0.0002 | 16 |
| Breech delivery | 27.40 (SD: 13.81) | (20.74, 34.05) | <.0001 | 19 |
| Severe preeclampsia/eclampsia | 27.09 (SD: 15.24) | (19.26, 34.93) | <.0001 | 17 |
| Vaginal exam in labor | 26.16 (SD: 17.78) | (17.59, 34.73) | <.0001 | 19 |
| Hemorrhage- atony & manual removal | 25.79 (SD: 13.11) | (19.48, 32.11) | <.0001 | 19 |
| Newborn resuscitation | 25.48 (SD: 7.82) | (21.46, 29.50) | <.0001 | 17 |
| Infection in labor | 23.26 (SD: 13.10) | (16.74, 29.77) | <.0001 | 18 |
| Examination of the newborn | 19.01 (SD: 22.05) | (7.68, 30.35) | 0.0026 | 17 |
| Assessment in labor | 13.05 (SD: 13.98) | (6.51, 19.59) | 0.0005 | 20 |

*N = 20; n differs by OSCE skills because a pre- or post-test was not available for some participants.

**Table 4. Direct observation in the clinical setting competencies and mean difference in pre- and post-assessment score.**

| Direct observation competencies | Mean Difference (post - pre) & SD | 95% CI for Mean Difference | p-value | n* |
|---|---|---|---|---|
| Respectful maternity care | 18.10 (SD: 18.56) | (9.42, 26.79) | 0.0003 | 20 |
| Using the partograph | 14.28 (SD: 9.30) | (9.66, 18.91) | <.0001 | 18 |
| Spontaneous vaginal delivery | 11.66 (SD: 8.09) | (7.87, 15.45) | <.0001 | 20 |
| Teamwork & communication | 10.48 (SD: 11.30) | (5.19, 15.77) | 0.0005 | 20 |
| Repair of 1st & 2nd degree perineal tears | 9.26 (SD: 11.91) | (2.07, 16.46) | 0.0159 | 13 |

*N = 20; n differs by direct observation competencies because a pre- or post-test was not available for some participants.

**Table 5. Clinical skills self-assessment pre- and post-program differences in confidence and competence.**

| | Confidence (n=20 pre-; n=10 post) | | | | Competence (n=18 pre-; n=9 post) | | | |
|---|---|---|---|---|---|---|---|---|
| | **Mean** | **Median** | **IQR** | **p-value** | **Mean** | **Median** | **IQR** | **p-value** |
| Overall pre- | 2.8523 | 2.9511 | 0.8878 | | 2.1450 | 2.1136 | 0.5448 | |
| Overall post- | 4.5898 | 4.7098 | 0.6618 | | 2.9205 | 2.9288 | 0.1343 | |
| Difference | 1.8035 | 1.4323 | 0.8531 | 0.002 | 0.7943 | 0.7922 | 0.2868 | 0.0039 |
| Competency 1: General Competencies pre- | 2.6789 | 2.6348 | 1.0190 | | 2.0126 | 1.9142 | 0.4924 | |
| Post- | 4.4288 | 4.5833 | 0.8055 | | 2.8491 | 2.9166 | 0.2777 | |
| Difference | 1.9160 | 1.6025 | 1.1363 | 0.002 | 0.8323 | 0.8611 | 0.3680 | 0.0078 |
| Competency 2: Pre-pregnancy and antenatal care pre- | 2.8592 | 3.0547 | 0.9196 | | 2.1500 | 2.1847 | 0.5949 | |
| Post- | 4.584 | 4.6904 | 0.7142 | | 2.9023 | 2.8705 | 0.1479 | 0.0039 |
| Difference | 1.792 | 1.4924 | 0.8801 | 0.002 | 0.7174 | 0.6785 | 0.3358 | |
| Competency 3: Care during labor and birth pre- | 3.1584 | 3.3679 | 1.0997 | | 2.3691 | 2.3883 | 0.9287 | |
| Post- | 4.7307 | 4.9444 | 0.3777 | | 2.9834 | 3.0000 | 0.0185 | |
| Difference | 1.5601 | 1.2701 | 0.962 | 0.002 | 0.6188 | 0.5733 | 0.5638 | 0.0078 |
| Competency 4: Ongoing care of women and newborns pre- | 2.6811 | 2.8240 | 1.3633 | | 2.0221 | 1.9213 | 0.5015 | |
| Post- | 4.6614 | 4.7137 | 0.5016 | | 2.9303 | 2.9687 | 0.1269 | |
| Difference | 2.0586 | 2.0606 | 0.8383 | 0.002 | 1.0252 | 1.0581 | 0.7859 | 0.0078 |

Wilcoxon signed-rank test examining the average difference in median pre-post scores.

pre- and post-program scores each increased. Competency category 4: Ongoing care of women and newborns had the largest increase with a median difference in score of 2.06 (IQR: 0.83, p=0.002) while the smallest increase in score was in the Competency category 3: Care during labor and birth (median: 1.27, IQR: 0.96, p=0.002).

*Competence*

Preceptor self-assessed competence in precepting was also statistically significant with average median scores increasing 0.79 points (IQR: 0.29, p=0.0039) between pre- and post-program (Table 5). Competency 4: Ongoing care of women and newborns had the largest difference in score pre- and post-program (median: 1.06, IQR: 0.79, p=0.0078) while Competency 3: Care during labor and birth had the smallest difference in scores pre- and post-program (median: 0.57, IQR: 0.56m p=0.0078).

**Precepting/teaching competency & confidence**

**Student evaluation of preceptor.** Student evaluations were only collected at baseline from one site (n=10). Some preceptors had multiple student evaluations with an average of 4.85 student evaluations per preceptor (Table 6). There was no statistically significant difference in median pre- to post-program student evaluations of preceptors (median difference: 0.0233, IQR: 0.5345, p=0.4316).

**Observed assessment of preceptor's competence in precepting.** Direct observation took place in one facility with n=9 participants. There was no statistically significant difference in median pre- to post-program observed competence difference (median: 0, IQR: 0.0003, p=0.375) (Table 7).

**Self-assessment of preceptor's competence in precepting.** For the self-assessment of preceptor competence in precepting, there was an overall median difference of 1.33 points (IQR: 0.79, p=0.0002) (Table 8). Across the 8 roles of a preceptor the largest median difference in pre- and post-program scores was in Role 8: Collaborator (median: 1.67, IQR: 1.0000, p=0.0002) and the smallest difference in pre- and post-program scores was in Role 3: Role Model (median: 0.56, IQR: 0.3333, p=0.002).

**Table 6. Student evaluations of preceptor.**

|  | Mean | Median | IQR | p-value | n (preceptors) | n (evaluations) |
|---|---|---|---|---|---|---|
| Pre-Program | 2.3649 | 2.3890 | 0.2640 |  | 10 | 1064 |
| Post-Program | 2.4932 | 2.5340 | 0.2864 |  | 10 | 33 |
| Difference | 0.1283 | 0.0233 | 0.5345 | 0.4316 |  |  |

*Wilcoxon signed-rank test examining the average difference in median pre-post scores.

**Table 7. Observed assessment of preceptor's competence in precepting.**

|  | Mean | Median | IQR | p-value | n |
|---|---|---|---|---|---|
| Pre-Program | 2.3213 | 2.2553 | 0.4363 |  | 9 |
| Post-Program | 2.3286 | 2.240 | 0.4360 |  | 9 |
| Difference | 0.0072 | 0 | 0.0003 | 0.375 |  |

*Wilcoxon signed-rank test examining the average difference in median pre-post scores.

**Table 8. Self-assessment pre- and post-program differences in Precepting competence.**

| Role | Competence (n=20 pre-; n=13 post) | | | |
|---|---|---|---|---|
|  | Mean | Median | IQR | p-value |
| Overall for 7 roles pre-program | 3.7653 | 3.7576 | 0.8720 |  |
| Overall for 7 roles post-program | 4.7909 | 4.86 | 0.2465 |  |
| Difference | 1.1492 | 1.3285 | 0.7988 | 0.0002 |
| Role 1: Caregiver pre- | 4.1483 | 4.25 | 0.8333 |  |
| Post- | 4.8718 | 5.00 | 0.1667 |  |
| Difference | 0.8487 | 0.60 | 0.8333 | 0.0002 |
| Role 2: Guide pre- | 3.75 | 3.75 | 1.50 |  |
| Post- | 4.8462 | 5.00 | 0.3333 |  |
| Difference | 1.2821 | 1.00 | 1.3333 | 0.0002 |
| Role 3: Role Model pre- | 4.2458 | 4.3889 | 0.4861 |  |
| Post- | 4.8639 | 4.8889 | 0.2222 |  |
| Difference (n=20; n=10) | 0.6889 | 0.5556 | 0.3333 | 0.002 |
| Role 4: Facilitator | 3.6771 | 3.60 | 0.7667 |  |
| Post- | 4.7695 | 4.7333 | 0.2424 |  |
| Difference | 1.1643 | 1.1333 | 0.8333 | 0.0002 |
| Role 5: Supervisor | 3.7999 | 3.8333 | 1.1667 |  |
| Post- | 4.6667 | 5.00 | 0.6667 |  |
| Difference (n=20, n=9) | 1.1481 | 1.00 | 1.00 | 0.0156 |
| Role 6: Evaluator pre- | 3.575 | 3.75 | 1.6875 |  |
| Post- | 4.8671 | 5.000 | 0.1429 |  |
| Difference (n=20; n=9) | 1.631 | 1.50 | 1.2679 | 0.0039 |
| Role 7: Collaborator pre- | 2.50 | 2.50 | 1.000 |  |
| Post | 4.3846 | 4.6667 | 1.000 |  |
| Difference | 1.9487 | 1.6667 | 1.000 | 0.0002 |

Wilcoxon signed-rank test examining the average difference in median pre-post scores.

## Discussion

This midwifery preceptor program achieved statistically significant improvements in preceptors' clinical and teaching competence and confidence in its first year. This low-dose, high-frequency program may be a useful intervention to strengthen midwifery education and the quality of midwifery care in a region with the highest burden of maternal mortality.

It is estimated that approximately one million more midwives are needed globally, the greatest need being in sub-Saharan Africa, and there is a significant shortage of preceptors to teach midwifery students in the clinical setting [4]. Currently, the International Confederation of Midwives recommends at least 60% of midwifery curriculum is composed of clinical hours, while 40% is classroom-based [27]. This necessitates a large number of clinically based midwives who are competent and confident to create an enabling learning environment in the clinical setting [28,29].

The approach to teaching in the clinical setting greatly differs from classroom teaching and from providing direct patient care. Clinical teaching requires a unique skill set that involves supporting students to apply theory to practice in real-life medical situations while maintaining safety and high quality of care. Literature has documented that preceptors need and want training and support in this role [30]. Investing in preceptor training has also been found to be essential in building the critical thinking skills and competency of midwifery students [31–34].

This analysis demonstrates an association between increased pre- to post-assessment scores, indicating improved clinical competency and confidence of preceptors. Strengthening clinical competencies is a necessary component to consider incorporating into preceptor training because there is evidence that some midwives in clinical practice lack essential midwifery competencies. A study looking at the quality of preceptors in the Democratic Republic of the Congo cited preceptors had all different levels of education and lacked competence [35]. Studies in Kenya and Ethiopia have found that insufficient in-service training for midwives has led to poor provision of quality emergency obstetric and new-born care (EmONC) [36–38]. A study looking specifically at Sierra Leone, found a similar deficiency in practicing midwives' clinical competencies. Additionally, a scoping review in Africa found that deficiencies in care, specifically in recognizing obstetric emergencies and responding to them, is one of the leading factors contributing to maternal deaths in the region [39]. These studies point to the critical need to improve the clinical competencies of midwives who provide the majority of care in Africa. They also highlight the importance of ensuring that any clinical midwife aiming to be a preceptor has strong clinical competencies prior to teaching any student.

Aside from developing strong clinical skills, the midwifery preceptor program focused on developing the teaching competencies of preceptors. Research has shown that preceptors' knowledge of teaching methodologies, expectations as a preceptor, and ability for feedback and evaluation is lacking in midwifery education [4]. This program showed mixed results in improving enrolled preceptors teaching competency and confidence. Direct observation of preceptors' teaching competency did not show evidence of increased competency. Direct observation only occurred at one of the study sites and the number was small (n=9), which is perhaps a reason why results found no improvement from pre-post in this assessment. Alternatively, the program may not have effectively improved preceptor competency. However, in self-assessment and student assessment of preceptors' teaching competency and confidence, there were statistically significant improvements pre- to post-assessment. Student assessments did not have statistically significant changes; yet, student assessments were high to begin with at the pre-test. Of note, all preceptors completed and graduated from the program, and our qualitative findings identified both barriers and facilitators to the program, with the majority of individuals involved (facilitators, participants, and administrators) finding the program to be organized and supportive of preceptors' needs [12].

Implementation of this preceptor program takes approximately six months and requires a substantial investment by both the preceptor participants as well as the facilitators. In a qualitative analysis of the program, participants stated it was a challenge balancing the workload of this program with other professional and personal responsibilities [12]. However, there is strong evidence that the low dose, high frequency approach using multiple techniques accompanied by mentorship or supportive supervision at clinical sites has been shown to result in greater clinical performance, knowledge and skills retention, and health outcomes than single, isolated interventions [15,40–45]. In a systematic review of interventions

to improve health care provider performance in low- and middle-income countries, the traditional one-off educational session approach has shown a low effectiveness [46].

To the authors' knowledge, many countries in the region are attempting to implement nursing and midwifery preceptor programs; these efforts often rely on traditional, one-off educational sessions, and there is limited rigorous evaluation to demonstrate their effectiveness or impact. The U.S. President's Emergency Plan for AIDS Relief, Nurse Education Partnership Initiative (NEPI) launched a clinical preceptor program in Zambia and Malawi in 2013; however, there is little published regarding the effectiveness and sustainability of these NEPI programs [47]. There is even less knowledge about how to develop and sustain such a program. The authors of this study have been involved in a similar nursing preceptor program in pediatrics in Malawi that has shown improvements in clinical and precepting confidence and competence [48].

The International Confederation of Midwives' (ICM) Global Standards for Midwifery has detailed requirements for clinical preceptors to maintain safety and high quality of midwifery education. The ICM states that clinical preceptors must "demonstrate competency in practice and maintain this competency," and have "principles of adult teaching and learning; skills in facilitating student inquiry and participation; ability to impart information; and the ability to evaluate student performance" [27]. Additionally, ICM states that there must be written documentation of each preceptors' maintenance of teaching and clinical competencies. By ensuring and documenting clinical and teaching competency, this midwifery preceptor program achieves many of the ICM requirements for clinical preceptors.

While this study focused on the program increasing clinical and teaching competency and confidence of preceptors, there remains ongoing evaluation to assess the impact of the program on midwifery student's competency and maternal and neonatal outcomes in facilities where the program is being implemented. Of note, the student evaluation scores were relatively high at pre-test, leaving little room for improvement, and there is cultural, and hierarchical aspects to be aware of when interpreting student evaluations of preceptors. Psychological safety in clinical settings have been evaluated globally, and it can be difficult to solicit honest feedback from students as they often are not forthcoming with negative remarks in case of retribution, power dynamics, or fear of reprisal [49]. Defining and increasing psychological safety is built into some modules of this preceptor program.

Like all studies, this is not without limitations. Assessments were completed by experienced midwives and midwifery teachers; like all observational assessments, there was some subjectivity to grading. However, it is well established that OSCEs (in particular), are a valid and reliable form of assessment for examining competency after participating in a healthcare education program [50–53]. The sample was relatively small, limiting the ability to conduct complex analyses and limiting generalizability. This program was piloted in two cities at two regional general hospitals and has yet to be applied in more remote or smaller settings. Additionally, not all assessments were completed by each participant. These assessments were time intensive, requiring approximately four to six weeks to complete testing with the facilitators (less time per participant, but still at least portions of multiple days to complete). Therefore, we did have substantial missing data and restricted our analyses only to participants who had completed both pre- and post-testing for the various assessments. Two self-reported tools were used to help triangulate self-reported confidence. Although there are known limitations of self-report such as social desirability in students, self-report is also known to be the only way to report self-reported outcomes [54,55]. Finally, alternative explanations for statistically significant findings could be biased due to phenomena such as the Hawthorne effect or maturation bias [56,57].

## Conclusion

Participants in this innovative midwifery preceptor program demonstrated statistically significant improvements in clinical and precepting competence and confidence in written and structured exams. No significant difference was observed in clinical observation nor student evaluations. While this pilot study was limited in scale, we observed a consistent positive increase across pre- to post-intervention assessments. These preliminary findings suggest that this low-dose, high-frequency preceptor program is a promising model for strengthening midwifery education in low resource settings. To build

on this proof-of-concept in two sites, future large-scale, multi-site studies are essential to establish the generalizability of these outcomes and to evaluate the long-term impact of these clinical competencies on maternal health outcomes.

## Author contributions

**Conceptualization:** Brittney J. van de Water, Ashley H. Longacre, Mustapha Sonnie, Frances Fornah, Patricia Juana-Kamara, Mary Augusta Mamakoh Fullah, Julie Mann.

**Data curation:** Brittney J. van de Water, Mary Moran, Jenny Hotchkiss, Isha Beckie Sesay.

**Formal analysis:** Ashley H. Longacre.

**Funding acquisition:** Brittney J. van de Water, Julie Mann.

**Investigation:** Brittney J. van de Water, Mustapha Sonnie, Jenny Hotchkiss, Marian Bangura, Chrisensia Owoko, Jennina Rose Viera, Alice Konyani, Adelaide Debrah, Julie Mann.

**Methodology:** Brittney J. van de Water, Ashley H. Longacre, Julie Mann.

**Project administration:** Brittney J. van de Water, Mustapha Sonnie, Marian Bangura, Isha Beckie Sesay, Frances Fornah, Patricia Juana-Kamara, Chrisensia Owoko, Jennina Rose Viera, Alice Konyani, Adelaide Debrah, Julie Mann.

**Resources:** Brittney J. van de Water, Julie Mann.

**Software:** Ashley H. Longacre.

**Supervision:** Brittney J. van de Water, Mustapha Sonnie, Marian Bangura, Isha Beckie Sesay, Frances Fornah, Patricia Juana-Kamara, Chrisensia Owoko, Jennina Rose Viera, Alice Konyani, Adelaide Debrah, Mary Augusta Mamakoh Fullah, Julie Mann.

**Validation:** Mary Moran, Jenny Hotchkiss, Marian Bangura, Chrisensia Owoko, Jennina Rose Viera, Alice Konyani, Adelaide Debrah, Julie Mann.

**Writing – original draft:** Brittney J. van de Water.

**Writing – review & editing:** Ashley H. Longacre, Mustapha Sonnie, Mary Moran, Jenny Hotchkiss, Marian Bangura, Isha Beckie Sesay, Frances Fornah, Patricia Juana-Kamara, Chrisensia Owoko, Jennina Rose Viera, Alice Konyani, Adelaide Debrah, Mary Augusta Mamakoh Fullah, Julie Mann.

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
