## [Decision Letter · Decision Letter 0]

23 Apr 2026

Effectiveness of a novel midwifery preceptor program on midwives’ competence and confidence in Sierra Leone: a prospective pilot study

PGPH-D-26-01122

Dear Dr. van de Water,

We are pleased to inform you that your manuscript 'Effectiveness of a novel midwifery preceptor program on midwives’ competence and confidence in Sierra Leone: a prospective pilot study' has been provisionally accepted for publication in PLOS Global Public Health.

Best regards,

Tanmay Bagade, Ph.D., MS (O&G), MPH, MHM

Academic Editor

Reviewer Comments (if any, and for reference):

Reviewer's Responses to Questions

Comments to the Author

1. Does this manuscript meet PLOS Global Public Health’s publication criteria? Is the manuscript technically sound, and do the data support the conclusions? The manuscript must describe methodologically and ethically rigorous research with conclusions that are appropriately drawn based on the data presented.

Reviewer #1: Yes

2. Has the statistical analysis been performed appropriately and rigorously?

Reviewer #1: Yes

3. Have the authors made all data underlying the findings in their manuscript fully available (please refer to the Data Availability Statement at the start of the manuscript PDF file)?

Reviewer #1: Yes

4. Is the manuscript presented in an intelligible fashion and written in standard English?

Reviewer #1: Yes

5. Review Comments to the Author

Reviewer #1: 1. Data supports conclusions.

2. The statistical analysis been performed appropriately and rigorously.

3. Findings are available.

4. Manuscript is written in English.

Additional Comments:

Consider strengthening the feasibility section. Some of the information in the Discussion regarding other nursing programs could compliment the background section, as well.

6. PLOS authors have the option to publish the peer review history of their article (what does this mean?). If published, this will include your full peer review and any attached files.

Do you want your identity to be public for this peer review? For information about this choice, including consent withdrawal, please see our Privacy Policy.

Reviewer #1:  Yes: Libby Marie Bunker
